# Clinical Performance of a Novel Two-Piece Abutment Concept: Results from a Prospective Study with a 1-Year Follow-Up

**DOI:** 10.3390/jcm10081594

**Published:** 2021-04-09

**Authors:** Giacomo Fabbri, Tristan Staas, Tomas Linkevicius, Valda Valantiejiene, Oscar González-Martin, Eric Rompen

**Affiliations:** 1Ban Mancini Fabbri Dental Clinic, 47841 Cattolica RN, Italy; 2Tandartspraktijk Staas & Bergmans, 5216 XA’s-Hertogenbosch, The Netherlands; t.staas@gmail.com; 3Implantology Center, 03162 Vilnius, Lithuania; linktomo@gmail.com; 4Dental Clinic Auksteja, 44275 Kaunas, Lithuania; valda.valantiejiene@gmail.com; 5Gonzalez+Solano Atelier Dental, Blanca de Navarra 10, 28010 Madrid, Spain; oscargm76@yahoo.es; 6Service de Médecine Dentaire, 4000 Liège, Belgium; erompen@hotmail.be

**Keywords:** two-piece abutment, soft tissue health, prospective

## Abstract

Development of a stable and healthy soft-tissue barrier around dental implants is key to long-term success of implant-supported prostheses. The novel two-piece abutment concept shifts the prosthetic interface to the soft-tissue level to protect bone interface/connective tissue during the healing phase and restorative procedures. This prospective study included 72 patients treated with 106 implants to support a single-tooth or a three-unit bridge restored with two-piece abutments. The evaluation included marginal bone level change (MBLC), implant and prosthetic survival, soft-tissue health including keratinized mucosa height and mucosal margin position, patient quality of life (QoL) and satisfaction, and clinician satisfaction and ease-of-use rating of the concept. Mean MBLC from implant placement to 1 year was −0.36 ± 1.26 mm (*n* = 89), the 1-year implant and prosthetic survival rates were 97.1 and 96.7%, respectively, while keratinized mucosa height increased from 2.9 ± 1.2 mm at prosthetic delivery to 3.2 ± 1.3 mm, and mucosal margin migrated coronally by 0.49 ± 0.61 mm by 1 year. Patient satisfaction and QoL were high. Clinicians were satisfied with the esthetic and functional results and rated the concept as easy to use. In conclusion, the novel two-piece abutment concept promotes good peri-implant tissue health, while providing an easy-to-use workflow and high treatment satisfaction to both patients and clinicians.

## 1. Introduction

Dental caries and tooth loss remain the most prevalent and consequential oral diseases with a global health and economic burden [1]. As such, necessity to treat these diseases has driven rapid development of technologies and materials applied in restorative dentistry. Modern dental materials such as lithium disilicate or zirconia have become the preferred choice due to their high esthetics and excellent mechanical properties [2]. Furthermore, current state-of-the-art restorative treatment methods for cavities in natural teeth and endodontic therapy consider not only the type of dental material and polymerization time but also the cavity angles and preparation techniques designed to decrease the risk of fracture of the restoration or tooth fracture, as well as use of cores and posts [3,4,5]. Similarly, technology has introduced new advances in implant dentistry. When assessing implant success, criteria additional to implant survival, such as stability of peri-implant hard and soft tissues and patient-centered outcomes, are also considered [6]. In particular, the amount of peri-implant bone loss following fixture placement and abutment connection is a major contributor to long-term implant success. Several factors including implant position, design, surface treatment, and connection geometry have been found to influence peri-implant hard tissue stability [7,8,9]. For example, the extent of early bone remodeling, i.e., the remodeling that takes place immediately after implant placement, depends on soft tissues thickness, collar surface and macro-design, implant drilling protocol, surgical trauma and possible infections, restorative materials, abutment surface/design, and its connection protocol [10,11,12,13,14,15,16,17].

The development of a stable and healthy soft tissue barrier around dental implants is an indispensable prerequisite for the long-term success of implant-supported prostheses [18]. However, healing of mucosa around a trans-mucosal implant component is challenging in the oral environment, due to the constant exposure to microorganisms. A rapid formation of an efficient soft tissue seal is, therefore, crucial to establish and maintain healthy peri-implant tissues. When a dental implant is placed and restored, soft tissues should adhere onto the abutment and restoration surfaces, preventing penetration of plaque and bacteria into the sulcus; this adherence represents a defense area that protects the bone interface and, therefore, the osseointegrated implant [18]. Every effort to minimize disturbances to the healing phase may prevent connective tissue micro-damages and apical migration of the epithelium with consequent bone loss, even when the biological width is already formed. Repeated connection and disconnection of abutments can compromise the stability of healing both in early stages (i.e., hemostasis, inflammatory and proliferation phases) and in the long term (i.e., remodeling phase) [19]. One strategy proposed as an effective approach to preserve peri-implant hard and soft tissues has been the use of definitive abutments placed at implant insertion and never removed thereafter [16,20]. A number of studies have tested this hypothesis and shown that multiple abutment disconnections associate with an increase in marginal bone loss [21,22,23]. Moreover, other clinical studies indicate favorable results when an approach to avoid abutment disconnections is applied [24,25,26,27]; however, these investigations differed in their surgical and clinical protocols, such as site type (healed sites versus post-extraction alveolar sockets), soft tissue thickness, vertical implant position, loading protocol, prosthetic retention, abutment design, implant design, and implant connection. In addition, the “one abutment one time” approach has the disadvantage of forcing the surgeon to anticipate the final soft tissue margin and, thus, to make the decision regarding the prosthetic finishing line at surgery.

Tissue level implants are another design developed to promote undisturbed soft tissue healing. However, these implants are mainly used in the posterior, because they offer suboptimal esthetics required in the anterior region [28].

In an effort to provide a system that protects the soft tissue seal from the time of implant placement and avoids the caveats of the ”one abutment one time” approach or tissue level implants, the novel On1 concept (Nobel Biocare AB, Göteborg, Sweden) was developed. This concept is based on a two-piece abutment, where the On1 base is placed on the implant at time of surgery and not disconnected thereafter. A provisional abutment, which is placed onto the On1 base at surgery, can easily be exchanged when the final abutment is placed. This approach shifts the prosthetic interface to the soft tissue level so that the bone interface and connective tissue are protected and preserved during the healing phase and all restorative procedures (Figure 1). The shift of the prosthetic interface to a more coronal position also simplifies clinical and technical procedures and at the same time reduces the patient’s discomfort during the therapy.

This prospective study was designed to test the hypothesis that the two-piece abutment concept would maintain stable bone levels from prosthetic delivery 12 weeks post implant insertion to 1 year post prosthetic delivery. In this manuscript, we present the clinical and radiological outcomes of the On1 concept, an evaluation of its impact on esthetics, quality of life, patient and clinician satisfaction, and an assessment of the ease-of-use of the concept.

## 2. Materials and Methods

### 2.1. Ethical Considerations

This multicenter clinical study, registered at clinicaltrials.gov (NCT03100448), was conducted according to the ethical principles set by the Declaration of Helsinki (as revised, amended, and clarified in October 2013). The investigators received ethics approvals for their respective study sites, and each subject gave his/her written informed consent to participate in the study.

### 2.2. Study Design, Setting, and Participants

This open prospective multi-center clinical trial was designed to evaluate bone response and soft tissue healing around implant sites restored according to the On1 treatment concept (Nobel Biocare AB, Göteborg, Sweden). Four European private clinics in Belgium, Lithuania, the Netherlands, and Italy participated in the study. The first patient visit took place on 2 February 2017, the first implant was placed on 24 February 2017, and the last follow-up visit was recorded on 8 May 2019.

The primary inclusion criteria were (1) written informed consent; (2) ≥ 18 years of age and ceased growth; (3) willingness and ability to comply with all study related procedures (such as exercising oral hygiene and attending all follow-up procedures); (4) overall good health and compliance with good oral hygiene; (5) full-mouth bleeding score (FMBS) lower than 25% and full-mouth plaque score (FMPI) lower than 20%; (6) favorable and stable occlusal relationship; (7) healed sites (i.e., minimum of 6 weeks post extraction) in need for and suitable for implant treatment in the posterior, pre-molar, and canine area in either jaw; (8) in need of one or multiple single tooth replacements or 3-unit bridges; (9) implant site free from infection and extraction remnants; (10) ability to undergo a 1-stage surgical procedure; (11) sufficient amount of buccal and lingual keratinized mucosa as per discretion of the treating clinician.

Patients were excluded due to (1) health condition that did not permit surgery or restorative procedure; (2) reason to believe that the treatment might have a negative effect on the patient’s overall situation (psychiatric problems), as noted in patient records or in patient health history; (3) any disorders in the planned implant area, such as previous tumors, chronic bone disease, or previous irradiation in the head/neck area; (4) an infection in the planned implantation site or adjacent tissue; (5) acute, untreated periodontitis in the planned implantation site or adjacent tissue; (6) ongoing application of interfering medication (steroid or bisphosphonate therapy, etc.); (7) uncontrolled diabetes, i.e., a patient with diagnosed diabetes that has a history of neglecting doctor’s recommendations regarding treatment, food, and alcohol intake or A1c level above 8%; (8) alcohol or drug abuse as noted in patient records or in patient health history; (9) heavy smoking (>10 cigarettes/day); (10) severe bruxism or other destructive habits; (11) pregnancy or breast-feeding at the time of implant insertion, (12) previous bone augmentation (lateral and/or vertical) at the implant site; (13) soft tissue augmentation less than 2 months before implant placement.

At the time of surgery, patient eligibility was reassessed. Patients were included if (1) they had a sufficient amount of bone for placing variable-thread tapered implants (NobelActive, Nobel Biocare AB) with a length of at least 8 mm; (2) their implants demonstrated primary stability and were able to receive the abutment with the required 35 Ncm torque.

### 2.3. Surgical and Prosthetic Protocol

The planned restorations included single crowns and 3-unit bridges. Healed implant sites received NobelActive implants (Nobel Biocare AB) according to the manufacturer’s instructions. A vertical distance of 3.0 to 3.5 mm from the implant/abutment interface to the free mucosal margin was considered to accommodate the biologic width in the soft tissues thickness; therefore, the implants were placed equi- or sub-crestally in relation to the supra-crestal soft tissue margin. Immediately after implant placement, the On1 base (Nobel Biocare AB) was connected with the implant using the On1 base clinical screw (Nobel Biocare AB). If proper seating of the base was not possible, a bone mill was used to remove the surrounding bone. Following On1 base placement, either an On1 healing cap or an On1 IOS healing cap (both Nobel Biocare AB) was attached to the On1 base for the healing period of ten weeks. Soft tissue grafting at surgery was allowed. After the surgery, patients were provided with home-care maintenance instructions and scheduled for post-surgical follow-up visits on an individual basis. Two weeks post implant placement, a digital impression using a 3Shape TRIOS intraoral scanner (3Shape A/S, Copenhagen, Denmark) according to the manufacturer’s instructions was taken. During the digital impression visit, all titanium healing caps were replaced with the IOS healing caps to allow for scan acquisition. Definitive prosthesis delivery was scheduled within 8–12 weeks after implant placement. At prosthetic delivery, the On1 base was retightened if necessary, and patients received On1 universal abutments, On1 esthetic abutments titanium, or On1 esthetic abutments zirconia (all Nobel Biocare AB). The final restorations were screwed or cemented using a phosphate or a resin cement.

### 2.4. Outcome Measures

The primary outcome measure was the change in the marginal bone level (ΔMBL) from prosthetic placement (baseline) to 12 months after the prosthetic placement. Secondary outcome measures included digital assessment of soft tissue thickness and clinical assessment of soft tissue health, as well as survival and success of implants and prosthetic restorations. Other assessed outcome measures were patient pain perception throughout the study period, patient oral health related quality of life, patient and clinician satisfaction with function and esthetics, and ease-of-use and the clinician’s confidence in the On1 concept. Marginal bone levels were assessed using standardized intraoral periapical radiographs taken at implant insertion, prosthesis placement, as well as 6 and 12 months after prosthesis placement. Radiographic examination was done using a standardized long-cone parallel technique using an X-ray holder customized to each patient in order to ensure a high quality comparable X-ray evaluation. Images were collected digitally or conventionally, and only those perpendicular to the implant, with a clear thread profile, and with at least 2 mm of visible surrounding bone, were used for the analysis. All bone-height measurements were made by an independent radiologist (University of Gothenburg, Sweden) using Adobe Illustrator and were defined as the distance between the reference point (implant platform) and the most apical level of the bone. Bone levels were recorded mesially and distally, and marginal bone levels were presented as averages, (mesial + distal)/2. Negative numbers indicate bone levels below the reference point and positive numbers indicate bone levels above the reference point. Marginal bone level changes were calculated for each side of the implant (mesial and distal) separately, as the difference between bone levels at two time points. The average of mesial and distal remodeling was then calculated for each implant site (paired for each side between two different time points). Negative numbers indicate bone loss. Missing data were not imputed and not included in evaluation.

Soft tissue health was assessed using the keratinized mucosa status, keratinized mucosa height, modified Plaque Index, the Gingival Index, and modified Sulcus Bleeding Index. Keratinized mucosa status was registered from implant insertion up to the 1-year follow-up visit, as 0 = no keratinized mucosa around the implant, 1 = mucosa surrounding the implant is partially keratinized, or 2 = the entire mucosa surrounding the implant is keratinized. The keratinized mucosa height at the implant buccal site was measured in mm from the attached mucosal margin to the junction with mobile mucosa, from implant insertion up to the 1-year follow-up visit. The modified Plaque Index (mPI) to evaluate plaque accumulation [29] was assessed at 3-, 6-, and 12-month follow-up visits, as 0 = no detectible plaque, 1 = plaque only recognized by running a probe across the marginal surface of the implant, 2 = plaque can be seen with the naked eye, 3 = abundance of soft matter. The gingival status was recorded at 3-, 6-, and 12-month follow-up visits using the Gingival Index (GI), a modified version of Löe and Silness (1963) classification [30], as 0 = normal gingiva surrounding crown/control tooth, 1 = mild inflammation, with slight color change and slight edema, 2 = moderate inflammation, with redness, edema and glazing, 3 = severe inflammation, with marked redness and edema, as well as tendency to spontaneous bleeding. The bleeding tendency was assessed at 3-, 6-, and 12-month follow-up visits by a modified Sulcus Bleeding Index (mBI) [29], as 0 = no bleeding when a periodontal probe is passed along the gingival margin adjacent to the implant, 1 = isolated bleeding spots visible, 2 = blood forms a confluent red line on the margin, 3 = heavy or profuse bleeding.

Soft tissue stability was assessed by comparing digital impressions collected with an intra-oral scanner at the prosthesis placement visit and the 12-month follow-up visit (the scans collected during the digital impression visit, which was two weeks after implant placement, were not included in the analysis due to post-surgical swelling of the soft tissue). The stl files obtained from the intraoral optical scan were uploaded to a digital shape sampling and processing software for re-elaboration of 3D models from the 3D scan data (Geomagic Qualify 12, 3D Systems, Rock Hill, SC, USA). For each patient, the two scans were superimposed and compared, based on a previously described procedure [31]. The scan acquired at prosthesis placement was set as reference, while the scan at the 12-month visit was set as test. For each superimposition, sagittal sections were obtained perpendicular to the axis of the prosthetic crown and one sagittal section corresponding to the mesio-distal center of the crown was selected. To assess the horizontal change in the coronal mucosal (gingival) margin position, the 2D linear horizontal distance between reference and test soft tissue profile was measured from 1 to 5 mm on the vestibular side, at 1 mm intervals, in an apical direction from the gingival margin. To assess the vertical change in the mucosal margin, the 3D models were superimposed by combining best match of manually selected surfaces and automated alignments, saved as a WRP file and compared using the same software package as for horizontal measurements.

Implant and prosthesis survival and success were assessed as follows. Cumulative survival rate (CSR) at implant level was calculated using a Kaplan–Meier survival analysis, in which only the withdrawn data were considered as censored. Implant success rate [32] was assessed at implant level from implant insertion to the 12-month follow-up, where a successful implant was defined as one that does not cause allergic, toxic or gross infectious reactions either locally or systemically, offers anchorage to a functional prosthesis, does not show signs of fracture or bending, does not show signs of peri-implant radiolucency on an intra-oral radiograph using a paralleling technique strictly perpendicular to the implant-bone interface, and does not show mobility when individually tested by tapping or rocking with a hand instrument. The final prosthesis survival and success status was registered at all follow-up visits. Final prostheses were classified as surviving if they were in situ, and as successful if they were in situ, and needed no repairs intra-orally or upon temporary removal. Prosthetic cumulative survival and success rates were calculated at prosthetic/restoration level using a Kaplan-Meier survival analysis, in which withdrawn restorations are assumed to follow the same trend regarding survival rate as the restorations that completed the study as per protocol.

Patient-reported outcome measures included pain assessment, oral health-related quality of life, and satisfaction with esthetics and function. Patients were asked to rate their pain during their visits from the post-surgical period up to prosthetic placement on a visual analogue scale (VAS) [33] from 0–10 where 0 = no pain to 10 = very intense. Patient quality of life was evaluated pretreatment, and at prosthetic placement, 6-month, and 1-year follow-up visits using the Oral Health Impact Profile (OHIP-14) [34]. The OHIP-14 questionnaires were made available in the respective local languages and validated translations and rated the prevalence of patients’ functional limitations, physical pain, psychological discomfort, physical, psychological and social disability, and handicap. The responses to the individual questions were scored on a 0–4 scale as follows: never = 0, hardly ever = 1, occasionally = 2, fairly often = 3, and very often = 4. Patients and clinicians rated their satisfaction with function and esthetics from the prosthetic placement visit up to the 1-year follow-up visit using a visual analogue scale [33] with ratings 1–10, where 10 = fully satisfied and 1= not satisfied. The ease of use of the On1 concept was assessed using questionnaires with visit-specific questions rated on a 0–10 scale. The three questions and their corresponding rating at implant insertion and On1 base placement visit were (1) How difficult was the placement of the On1 base on the implant? rated from 0 = difficult to 10 = very easy; (2) How would you rate the tactile feeling of the connection between the implant and the On1 base? rated from 0 = poor to 10 = excellent; and (3) Please rate the ease of use of the pre-mounted handle of the On1 base on a scale from 0 = poor to 10 = excellent. At the digital impression visit, the question was (1) How was the ease of use for taking impressions with the On1 impression coping on tissue level? rated from 0 = poor to 10 = excellent. Finally, at the prosthetic placement visit, 2 questions were asked: (1) How would you rate the tactile feeling of the connection between the On1 base and the prosthesis/post? Please rate on a scale from 1–10 rated from 0 = poor to 10 = excellent; and (2) The overall prosthetic placement procedure for the On1 concept was rated from 0 = poor to 10 = excellent. The confidence in the On1 concept were assessed at 3, 6, and 12 months by using a rating of 1–10, where 1 = very low confidence to 10 = very high confidence. Adverse events were monitored during the entire study period and registered in the electronic data capture system.

### 2.5. Statistical Analysis

Sample size calculation was performed for a single-arm study with a 1-year mean ΔMBL of −0.58 mm based on the weighted mean ΔMBL from 9 studies with early loading [35,36,37,38,39,40,41,42,43,44] compared to a historic control reference ΔMBL of −0.14 mm from a study with the same implant and with final abutments placed at surgery [45]. With the two-sided *t*-test at a significance level of α = 0.05, a power of 90%, and compensation for 20% subject withdrawal, a total of 68 subjects (17 subjects per study center) were to be included. The statistical evaluation considered all collected data from surgery and follow-up procedures. Missing data were not imputed and not included in the statistical evaluation. The distribution of continuous variables was given as mean and standard deviation (SD), and for categorical variables as frequency and percentage. Statistical comparison of score distribution for keratinized mucosa presence and height, mPI, GI, mBI, and soft tissue profile linear measurements at different time points were performed using the sign test. Center effects were estimated by means of univariate analysis of variance, with dependent variable being either continuous (age, marginal bone remodeling), nominal (gender) or ordinal (bone quality). All statistical analyses were performed using SPSS Statistics (version 25, SPSS Inc., Chicago, IL, USA).

This manuscript follows the STROBE guidelines.

## 3. Results

### 3.1. Patient Enrollment and Follow-Up

Based on primary inclusion and exclusion criteria, 86 patients were found eligible, signed the informed consent, and agreed to participate in this clinical study. After assessing the final inclusion criteria at the time of surgery 72 patients who received 106 implants were included. The baseline characteristics of the patient population as well as implants, implant sites, and surgical details are listed in Table 1. All patients attended the digital impression visit, which took place 14.5 ± 6.2 days (range, 6–64 days) after implant insertion. Two implants in two patients failed due to loss of stability, and no digital impression was taken for these implants. Subsequently, one patient with 4 implants was withdrawn from the study because they were not able to attend the remaining study visits, and two implants in one patient were withdrawn before the prosthetic delivery due to non-compliance with the study protocol (removal of the On1 base); the latter patient remained in the study, having three additional study implants. Overall, 69 patients (98 implants) received definitive prostheses, which were delivered 13.2 ± 6.7 weeks (range, 8–38 weeks) after implant insertion. The 3-, 6-, and 12-month follow-up visits were attended by 67 patients (95 implants), 66 patients (94 implants), and 67 patients (95 implants), respectively. A sample clinical case is shown in Figure 2.

### 3.2. Digital Impression and Prosthetic Delivery

The majority of implants (73.6%) were restored with a 1.75 mm high base, while the remaining implants received a 2.5 mm high base. The healing caps were an On1 titanium healing cap (84.0%) or an On1 IOS healing cap (16.0%). At the digital impression visit, the titanium healing caps were disconnected from the base, and the base needed to be retightened in 21 out of 87 cases. At the final prosthesis delivery, the bases needed to be retightened in 32 out of 98 cases. The final abutment type was mainly the On1 universal abutment (89.8%); the remaining abutments were the On1 esthetic abutment titanium (9.2%) and On1 esthetic abutment zirconia (1.0%). Eighty-three implants (84.7%) received a single crown restoration, while a three-unit bridge was placed on 15 (15.3%) implants. Most of the restorations (90.8%) were screw-retained and were all placed on the On1 universal abutment, while the cemented restorations were placed on the On1 esthetic abutments.

### 3.3. Outcome Measures

Analysis of the peri-implant marginal bone levels and remodeling revealed that the expected bone loss after implant insertion was followed by bone gain and, subsequently, stabilization of the bone levels. The mean marginal bone levels were −0.40 ± 0.95 mm (*n* = 100) at implant surgery, −1.02 ± 0.85 mm (*n* = 98) at prosthetic delivery, −0.83 ± 0.72 mm (*n* = 91) six months after the prosthesis placement, and −0.78 ± 0.88 mm (*n* = 95) twelve months after the prosthesis placement (Figure 3). Mean ΔMBL from implant surgery to prosthetic placement, 6-, and 12-month follow-up was −0.60 ± 1.16 mm (*n* = 93), −0.40 ± 1.15 (*n* = 85), and −0.36 ± 1.26 mm (*n* = 89), respectively. Mean ΔMBL from prosthetic placement to 6- and 12-month follow-up was 0.22 ± 0.61 mm (*n* = 89) and 0.23 ± 0.76 mm (*n* = 93), respectively. The primary outcome measure, ΔMBL from prosthetic placement to the 12-month follow-up, confirmed the hypothesis of stable marginal bone levels for the novel two-piece abutment concept used in the study.

Soft tissue health reflected the good hard tissue outcomes (Table 2 and Figure 4). The vast majority (over 90%) of study implants were surrounded by keratinized mucosa from surgery to the last follow-up at 12 months post final prosthesis delivery (*p* > 0.05 for all in-between group tests). The height of the keratinized mucosa decreased from surgery to prosthetic placement (*p* < 0.001), then showed a statistically significant increase by the 6-month follow-up (*p* = 0.026), and remained stable until the last follow-up at 12 months (*p* = 0.651). Increased plaque accumulation was evident from the 3- to the 12-month follow-up, while the gingival index worsened only minimally; however, the index changes between the different time points were not statistically significant. By contrast, the sulcus bleeding tendency continued to improve from the 3- to the 12-month follow-up, although the changes were not statistically significant. Finally, comparison of soft tissue profiles acquired with IOS at final prosthesis delivery with those at the 12-month follow-up showed that soft tissue remained stable (Figure 4). The vertical linear measurement shows a mean change of −0.49 ± 0.61 mm, indicating a coronal migration of the mucosal margin over the 1-year follow-up period. The sagittal linear measurements indicate a soft tissue profile increase in the first 2 mm and a slight decrease towards 4 and 5 mm apically from the gingival margin.

Out of the total of 106 implants, three implants failed; two failures were reported at the 2-week post-surgery digital impression visit, and one failure at the 3-month follow-up. Kaplan–Meier survival analysis showed a CSR of 98.1% at digital impression and prosthetic delivery, and 97.1% at 3, 6, and 12 months. Implant success rate was 98.1% at digital impression and prosthetic delivery, 97.1% at 3 and 6 months, and 95% at 12 months, the latter being due to two implants with signs of peri-implant radiolucency on the intra-oral radiographs. Out of the 91 prostheses placed, three failed the 3-month follow-up visit, yielding a prosthetic cumulative survival rate of 96.7% at the 3, 6, and 12 months. The prosthetic cumulative success rate was 95.4% at 3 months and 93.1% at 6 and 12 months.

Overall, patient-reported outcomes indicated that the procedure was associated with little pain (score of ≤ 1 on a scale of 0 to 10 for all timepoints; Figure 5a), significantly improved patient oral health-related quality of life from (Figure 5b), and resulted in very high satisfaction with both esthetics and function (score of ≥9.3 on a scale of 0 to 10 for all timepoints; Figure 5c,d). Clinicians rated their functional and esthetic satisfaction high, but slightly lower than their patients (score of 8.7 on a scale of 0 to 10 for all timepoints for function, score of ≥ 7.5 on a scale of 0 to 10 for all timepoints for esthetics; Figure 5c,d). Ease-of-use assessment results were very positive, except for the lukewarm 5.9 rating received by the pre-mounted base handle (Figure 5e). Overall, the clinicians’ confidence in the On1 concept was high (≥8 on a scale of 1 to 10) at all assessed time points (Figure 4f).

To evaluate whether treatment in different centers participating in this study resulted in different outcomes, the centers were compared for patient demographics including gender, age, and bone quality, as well as the primary endpoint, i.e., marginal bone remodeling. While the age of the treated patient population was comparable between all centers, the gender and bone quality distributions were significantly different (*p* = 0.02 and *p* = 0.003, respectively). Regarding marginal bone remodeling from implant placement to last follow-up and from final prosthesis delivery to last follow-up, there was no statistically significant difference between the participating centers.

### 3.4. Adverse Events

In total, 15 adverse device effects have been reported: nine technical (prosthetic complications) and six biologic (pain in four cases, implant loss in two cases). Fifteen additional adverse events, including three serious ones that resulted in hospitalization, occurred, none of which, however, were related to the study devices or the study procedure.

## 4. Discussion

The aim of this prospective clinical study was to assess clinical and radiological outcomes of the novel two-piece abutment concept applied to single tooth restorations and short-span bridges. The study has also evaluated patient-reported outcome measures as well as clinician feedback on the concept, including its ease of use.

The primary endpoint analysis of the present study demonstrated a statistically significant bone gain of 0.23 ± 0.76 mm from prosthetic delivery (at 12 weeks post implant placement) to the 12-month follow-up. This result is in contrast to the results of the reference study, in which the mean marginal bone remodeling was −0.14 ± 0.59 mm from final prosthetic delivery (at 3 months post implant placement) and a mean follow-up of 15.2 ± 4.8 months (range 12–36 months) [45], indicating that the novel two-piece abutment concept has beneficial effects on the marginal bone response. It is, nevertheless, important to note that the reference study, while it used the same study implant and applied the “one abutment one time” concept, it subjected the implants to immediate loading, which is in contrast to the current study. Importantly, the improved response with the On1 concept occurred with the absence of immediate loading, which was applied in the reference study, and is generally believed to have a positive impact on peri-implant bone response [46]. The evident bone gain is also a clear sign of excellent recovery, even in those cases that required the use of bone mill.

Only one study including restorations with the On1 two-piece abutment has been published to date [47]. This retrospective investigation compared the clinical outcomes of implants restored with a healing abutment versus implants restored according to the “one abutment one time” concept, including but not limited to the On1 two-piece abutment. The estimated mean bone loss in the latter group was 0.934 and 0.739 mm, depending on the implant type, with the mean follow-up of 20 months, which is more than the bone loss of 0.36 mm observed in the current study at 15 months post implant insertion (and 12 months post prosthetic delivery). It is not clear, however, whether these differences can be attributed to the different implant types used in the two studies, the longer follow-up in the published study, or the heterogenous group combing the On1 and the multi-unit abutment in the published study.

One likely factor that has resulted in the healthy bone response observed in the current study is the lack of multiple abutment reconnection and disconnection. Such improvement has been recently reported by systematic review and meta-analysis, in which a “one abutment one time” approach led to significantly smaller marginal bone loss in comparison to an approach requiring multiple abutment reconnections, even when the reconnection was performed only once or twice over the course of treatment [48].

Another factor that may have mitigated bone loss after implantation observed in the current study is sufficient vertical mucosal tissue width. Previously, implant sites with ≥3 mm of mucosal tissue have shown smaller marginal bone loss compared to those with medium and thin tissue, and patients with thinner soft tissue have been recommended to undergo a subgingival (3 mm below the gingival margin) implant placement [49]. A benefit of sub-crestal implant placement in thin soft tissue phenotypes on marginal bone response has also been demonstrated [17]. The subgingival/sub-crestal approach was applied in the current study and is likely to have positively contributed to the observed low marginal bone remodeling.

Analysis of soft tissue outcomes in the current study indicates excellent soft tissue health. Comparison of soft tissue profile from the prosthetic placement to the 12-month follow-up revealed a mean −0.49 ± 0.61 mm migration of the gingival margin, demonstrating overall stability of the peri-implant soft tissue. Similarly, the mean keratinized mucosa height and the bleeding index were stable from prosthetic delivery to the 12-month follow-up, with the latter showing a trend toward improvement. The excellent soft tissue health was recorded despite increased plaque accumulation and a trend toward decreased gingival index. These findings are consistent with the slight, although not statistically significant, improvements in soft tissue health reported in RCTs comparing the “one abutment one time” approach to the conventional protocols [48].

The implant level CSR in the current study was 97.1% (three implant failures in three subjects) at 12 months, which is consistent with the 95.7% to 100% range reported in the seven studies with early or delayed loading of the variable-thread tapered implant in rehabilitation of single teeth or partial edentulism and a follow-up of at least 1 year [50,51,52,53,54,55,56].

Statistical comparison of marginal bone remodeling across the different study centers revealed that there was no significant center effect, indicating the versatility of the novel On1 concept in that its performance does not appear to depend on the treating clinician.

Patient-reported outcomes have continued to gain importance in the assessment of oral care quality. Previous studies have shown that in general replacement of missing teeth with dental implants is met with a high functional and esthetic satisfaction in comparison to treatment with removable dentures [57]. However, how satisfied the patients are with their implant therapy depends on many factors, including patient expectations prior to therapy. The current study shows a very high patient satisfaction with both function and esthetics, a significant improvement in oral related quality of life, and minimal pain reported at surgery. These results reflect not only the excellent clinical and radiological outcomes observed in the study, including those for restorations in the highly challenging esthetic region, but are also likely to associate with the shift of the restorative work to the tooth level owing to the On1 base placed at surgery and not removed thereafter.

In general, the clinicians rated the novel On1 concept very positively. They were satisfied with the functional and esthetic outcomes and overall found the concept as easy to use. The pre-mounted handle was the sole component that had received a poor score; however, the handle has since been modified by the manufacturer to improve its ease-of-use (Nobel Biocare Services AG, personal communication). Nevertheless, the authors agree that there exists a learning curve with respect to the clinical use of this concept, such as application of the bone mill or the final implant position, which should facilitate optimal base placement.

The main limitation of the current study is its short follow-up. The results observed at the 12-month post prosthetic delivery indicate healthy hard and soft tissue responses, and future studies should address how these findings translate in the long term. Another limitation is the lack of direct comparison to the conventional approach or the “one abutment one time” approach; although the results of the current study fare well against those reported on the variable-thread tapered implants in similar indications, or the studies with the final abutment placed at surgery, only an RCT approach can compare these protocols directly.

## 5. Conclusions

Within the limitations of the 12-month follow-up, the novel two-piece abutment concept results in statistically significant marginal bone gain following the expected remodeling immediately after implant placement and promotes excellent soft tissue health. The concept shifts the restorative work above the soft tissue margin thereby allowing undisturbed soft tissue healing and better visibility for the restorative procedures. Its workflow is easy to use and provides high functional and esthetic satisfaction to both clinicians and their patients.

## Figures and Tables

**Figure 1 jcm-10-01594-f001:**
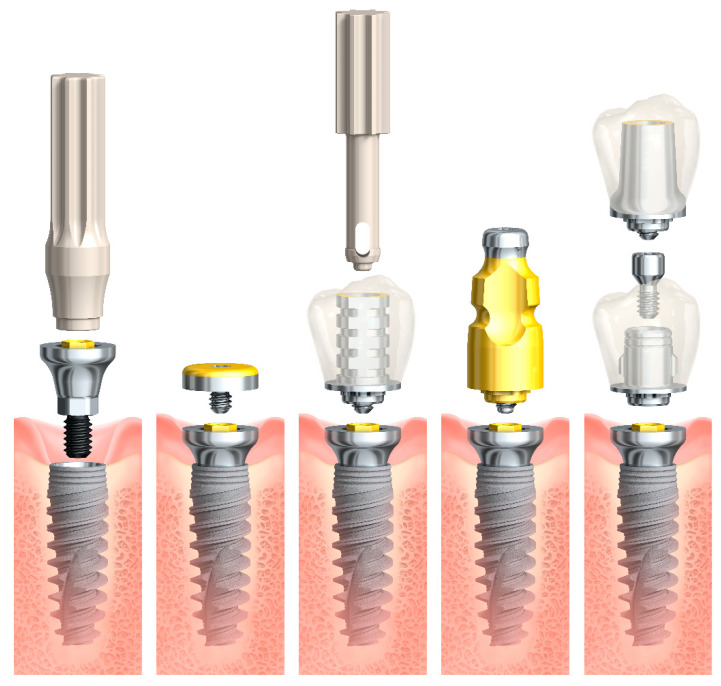
Prosthetic workflow of the On1 concept. From left to right: placement of the On1 base, healing cap, temporary prosthesis with temporary abutment, impression coping, and final prosthesis with either the On1 esthetic abutment or universal abutment (image courtesy of Nobel Biocare).

**Figure 2 jcm-10-01594-f002:**
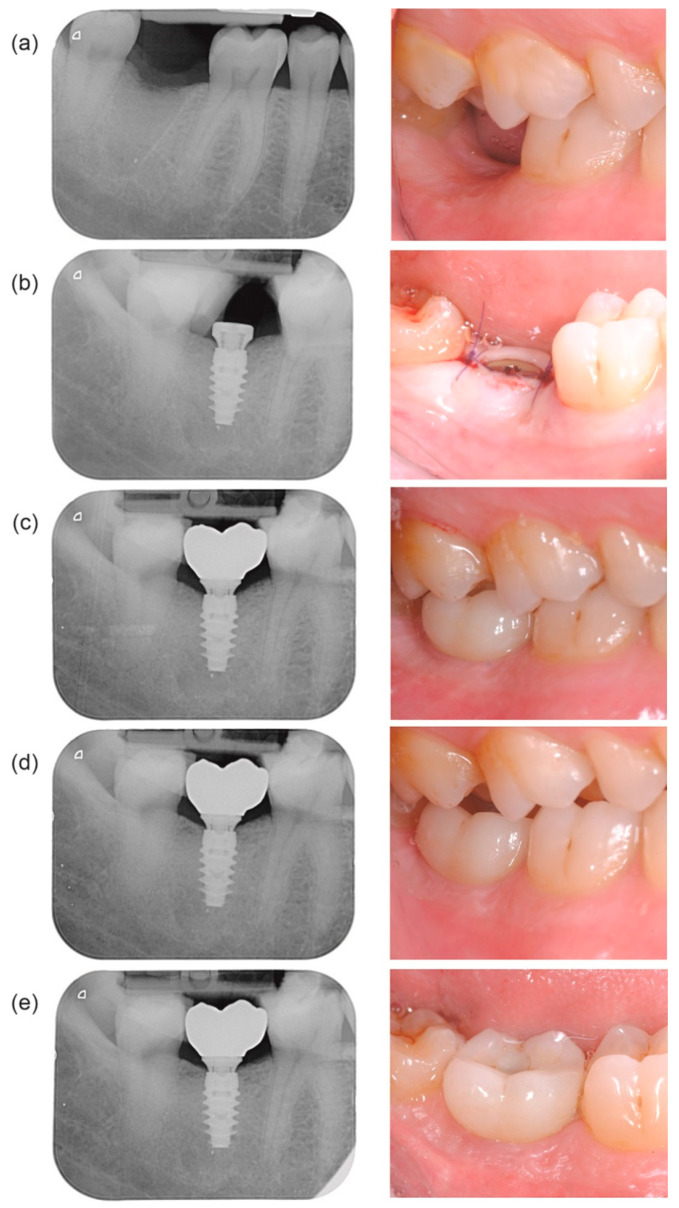
Sample clinical case from the study. Apical radiographs and clinical views acquired pre-surgery (**a**), at implant placement (**b**), at final prosthesis delivery ((**c**), left panel), 3 months later ((**c**), right panel), and 6 and 12 months after prosthetic delivery ((**d**,**e**), respectively).

**Figure 3 jcm-10-01594-f003:**
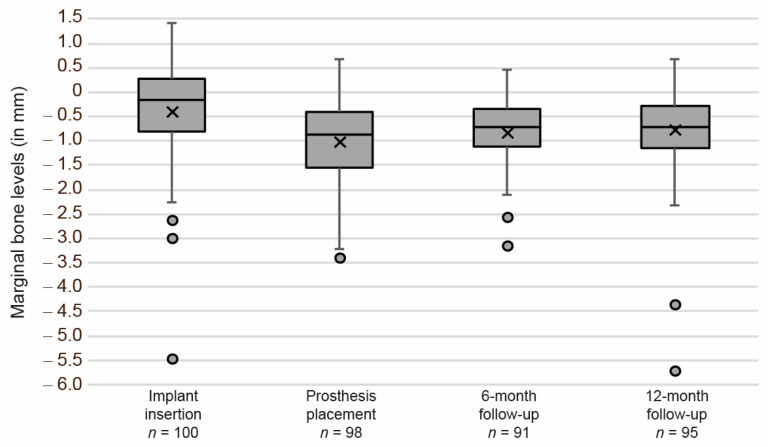
Evolution of marginal bone levels from implant insertion to the final follow-up, 12 months after prosthetic delivery. Box and whisker plot; circles indicate outliers and crosses the mean value.

**Figure 4 jcm-10-01594-f004:**
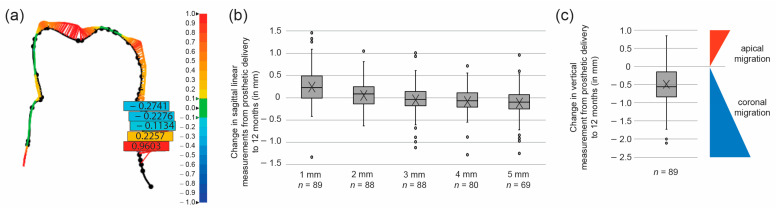
Soft tissue stability from prosthetic delivery to 12-month follow-up based on IOS profile analysis. (**a**) Example of a two-dimensional comparison of the digital profiles collected at the final prosthesis delivery and 12 months thereafter, with the five locations on the vestibular side chosen for linear measurements, from 1 to 5 mm in apical direction from the mucosal margin. Changes in volume are color coded, where green represents perfect alignment, the color range from yellow to red represents increasing size, and the color range from light to dark blue signifies size decrease (see scale on the right side of the panel). (**b**) Box and whisker plot illustrating the changes in the linear measurements at the five sagittal locations. Crosses indicate means and circles outliers. The number for sagittal measurements at 4 and 5 mm is lower because the scanner could not evaluate all the scan images in the most apical sagittal region. (**c**) Box and whisker plot illustrating the changes in the vertical measurement. Crosses indicate means and circles outliers. Note that the negative value reflects a coronal migration of the mucosal margin, i.e., a positive clinical outcome.

**Figure 5 jcm-10-01594-f005:**
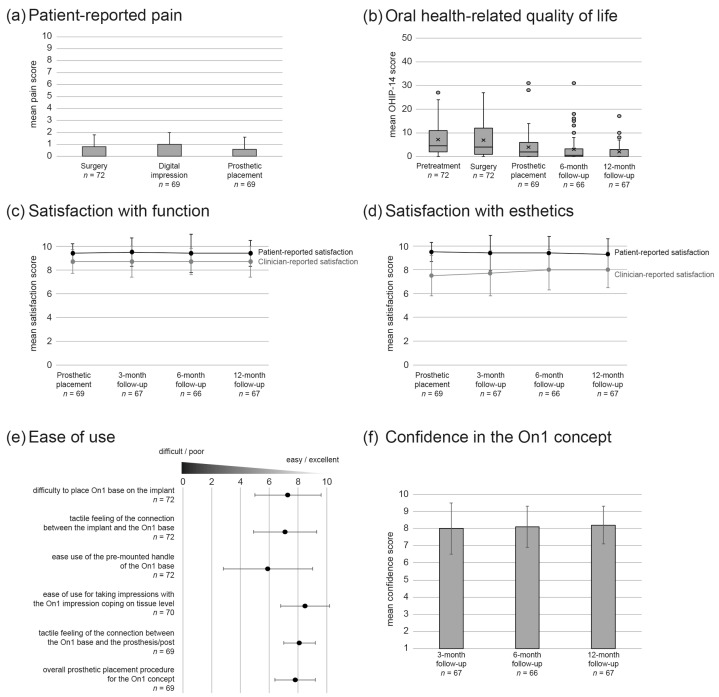
Patient-reported outcomes and clinician evaluation over the study period. For each group, the number of assessed sites is indicated under the group name. (**a**) Mean patient-reported pain score by follow-up. Error bars indicate standard deviation. (**b**) Box-and-whisker plot of the oral health-related quality of life according to Oral Health Impact Profile (OHIP-14) score. Decrease in OHIP-14 score reflects improved quality of life. Circles indicate outliers and crosses the mean score. (**c**,**d**) Mean patient and clinician satisfaction with function (**c**) and esthetics (**d**) by follow-up. Error bars indicate standard deviation. (**e**) Clinician-reported ease-of-use assessment. Error bars indicate standard deviation. (**f**) Clinician rating of confidence in the On1 concept by follow-up. Error bars indicate standard deviation.

**Table 1 jcm-10-01594-t001:** Baseline characteristics at surgery.

**Patient Characteristics**		***n* (%)**
Gender	Female	31 (43%)
	Male	41 (57%)
Age (mean ± SD, range) (years)		53.2 ± 12.7 (28–76)
Smoking status	Non-smoking	64 (88.9%)
	Smoking, ≤5 per day	3 (4.2%)
	Smoking, 6–10 per day	5 (7.0%)
History of periodontitis	Yes	12 (16.7%)
	No	60 (83.3%)
Implants per patient	1	49 (68.1%)
	2	17 (23.6%)
	3	3 (4.2%)
	4	1 (1.4%)
	5	2 (2.8%)
**Implant Characteristics**		***n* (%)**
Implant diameter (mm)	3.5	19 (17.9%)
	4.3	58 (54.7%)
	5.0	20 (18.9%)
	5.5	9 (8.5%)
Implant length (mm)	8.5	15 (14.2%)
	10.0	55 (51.9%)
	11.5	32 (30.2%)
	13.0	4 (3.8%)
**Implant Site/Surgical Characteristics**		***n* (%)**
Location	Maxilla	37 (34.9%)
	Premolars	20 (18.9%)
	Molars	17 (16.0%))
	Mandible	69 (65.1%)
	Premolars	13 (12.3%)
	Molars	56 (52.8%)
Bone quality	Very dense bone	10 (9.4%)
	Dense bone	40 (37.7%)
	Soft bone	45 (42.5%)
	Very soft bone	11 (10.4%)
Bone quantity	A	43 (40.6%)
	B	46 (43.4%)
	C	17 (16.0%)
	D, E	0
Bone mill use	Yes	57 (53.8%)
	No	49 (46.2%)
Soft tissue grafting	Yes ^1^	8 (7.5%)
	Advanced platelet-rich fibrin	3 (2.8%)
	Free gingival graft	5 (4.7%)
	No	98 (92.5%)
Insertion torque (mean ± SD, range) (Ncm)		49.6 ± 11.8 (35–70)

^1^ All soft tissue grafting was performed in a single participating center. SD, standard deviation; Ncm, Newton centimeter.

**Table 2 jcm-10-01594-t002:** Soft tissue health throughout the study.

Parameter	Implant Insertion	Prosthesis Placement	3-Month Follow-Up	6-Month Follow-Up	12-Month Follow-Up
*n*	106	98	95	94	95
**Keratinized mucosa status**					
No keratinized mucosa around the implant	-	-	1	-	1
Mucosa surrounding the implant is partially keratinized	7	3	5	3	2
The entire mucosa surrounding the implant is keratinized	99	95	89	91	92
**Keratinized mucosa height**					
Mean ± SD (mm)	3.5 ± 1.1	2.9 ± 1.2	3.1 ± 1.3	3.3 ± 1.4	3.2 ± 1.3
**Modified Plaque Index**					
No detectable plaque	not assessed	not assessed	66	56	44
Plaque only recognized by running a probe across the marginal surface of the implant			28	35	45
Plaque can be seen with the naked eye			1	3	6
Abundance of soft matter			-	-	-
**Modified Bleeding Index**					
No bleeding when a periodontal probe is passed along the gingival margin adjacent to the implant	not assessed	not assessed	68	71	76
Isolated bleeding spots visible			25	17	15
Blood forms a confluent red line on the margin			-	6	4
Heavy or profuse bleeding			2	-	-
**Gingival Index**					
Normal gingiva surrounding crown/control tooth	not assessed	not assessed	78	70	70
Mild inflammation—slight color change, slight edema			15	21	22
Moderate inflammation—redness, edema, and glazing			2	3	3
Severe inflammation—marked redness and edema, tendency to spontaneous bleeding			-	-	-

SD: standard deviation.

## Data Availability

Restrictions apply to the availability of these data. Data were obtained from Nobel Biocare Services AG and are available with the permission Nobel Biocare Services AG.

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
