# Peer review of "Clinical Performance of a Novel Two-Piece Abutment Concept: Results from a Prospective Study with a 1-Year Follow-Up"

_jcm, 2021, doi:10.3390/jcm10081594_

Round 1
Reviewer 1 Report
A clearly present paper describing a 2 stage abutment to enhanbce soft tissue adaptation and integration.
Reviewer 2 Report
- The approach is interesting and the topic is appropriate for the journal.
- The work has a clear structure and all the sections are well written in a way that is easy to read and understand.
- However, some modifications and improvements are needed to enhance the quality of the paper.
- The paper deals with the clinical performance of a novel two-piece abutment concept, reporting results from a prospective study with a 1-year follow-up. In the Introduction section, the authors start to discuss about dental implants. Accordingly, in the introduction section, the authors start to write: “Survival of dental implants is largely defined by the achievement and maintenance of osseointegration. When assessing implant success, additional specific criteria, such as stability of peri‐implant hard and soft tissues and patient‐centered outcomes, are also considered …”. However, even though the authors mainly focus on dental implants and clinical performance of a novel two-piece abutment concept, I also suggest to BRIEFLY improve the introduction starting from the several problems in dentistry and the adopted technical solutions (e.g., Dental Materials, 2017, 33(6), pp. 690–701; Dental Materials, 2017, 33(1), pp. e39–e47; Dental Materials, 2017, 33(12), pp. 1456–1465; Dental Materials, 2017, 33(12), pp. 1466–1472; Dental Materials, 2018, 34(7), pp. 1063–1071), and then focusing on the different concept related to the dental implants. All of this should improve the quality of the paper, reporting important features as well as different methodologies and technical solutions adopted for different cases in dentistry, thus helping the different kinds of readers to better understand the topic and the value of their work.
- The introduction and the list of references should be improved according to the above reported comments.
- The title is adequate and appropriate for the content of the article.
- The abstract contains information of the article.
- Figures and captions are essential and clearly reported.
Reviewer 3 Report
This is a well written paper that provides plenty of details of the cohort, methodology and evaluation techniques. The authors acknowledge funding from the maker of the implant but the data appears to stand on its own merit and the funding should not detract from the publication of results. The multiple citation to the maker can be reduced.
